# Extension of DBSCAN in Online Clustering: An Approach Based on Three-Layer Granular Models

**Xinhui Zhang** [1,†], **Xun Shen** [2,†] and **Tinghui Ouyang** [3,*]

1   Department of Journalism and Media, Nihon University, Tokyo 102-0074, Japan
2   School of Engineering, Tokyo Institute of Technology, Tokyo 152-8550, Japan
3   National Institute of Advanced Industrial Science and Technology, Tokyo 135-0064, Japan
*   Correspondence: ouyang.tinghui@aist.go.jp
†   These authors contributed equally to this work.

**Abstract:** In big data analysis, conventional clustering algorithms have limitations to deal with nonlinear spatial datasets, e.g., low accuracy and high computation cost. Aiming at these problems, this paper proposed a new DBSCAN extension algorithm for online clustering, which consists of three layers, considering DBSCAN, granular computing (GrC), and fuzzy rule-based modeling. Firstly, making use of DBSCAN algorithms' advantages at extracting structural information, spatial data are clustered via DBSCAN into structural clusters, which are subsequently described by structural information granules (IG) via GrC. Secondly, based on the structural IGs, a series of granular models are constructed in the medium space, and utilized to form fuzzy rules to guide clustering on spatial data. Finally, with the help of structural IGs and granular rules, a rule-based modeling method is constructed in the output space for online clustering. Experiments on a synthetic toy dataset and a typical spatial dataset are implemented in this paper. Numerical results validate the feasibility to the proposed method in online spatial data clustering. Moreover, comparative studies with conventional methods and existing DBSCAN variants demonstrate the superiorities of the proposed method, as well as accuracy improvement and computation overhead reduction.

**Keywords:** online clustering; DBSCAN extension; granular computing; three-layer model





## 1. Introduction

In today's information age, the most important task is to process huge information and data. Data mining is the basic tool in data analysis, which extracts interesting hidden patterns or features from data sources, and uses them in decision-making or future studies [1,2]. Among quantities of computational intelligence activities, clustering plays a fundamental and essential role in extensive applications, since the overwhelming majority of data are unlabeled nowadays. Clustering algorithms' major purpose is to group analogous objects into clusters by some intrinsic properties between data points [3], e.g., structural, statistical, distance-based, and density-based similarity, and so on. The resulted clusters are required to have maximum-similarity intra clusters, and a minimum of one inter cluster [4]. Due to these specialties, clustering results are usually utilized as the preprocessed step in further data mining studies, such as classification, prediction, anomaly detection, correlation analysis, etc. [5]. Therefore, to extract effective information in datasets efficiently, it is necessary and important to find a suitable clustering algorithm in data mining analysis.

Clustering algorithms have been studied and developed for the last several decades, as well as applied in real engineering applications, e.g., pattern recognition, scientific and social research, economic analysis, industrial engineering, biomedical data analysis, and so on [6,7]. According to the application preference, available data types, and particular purposes or requirements, clustering algorithms are mainly divided into five categories [8]: hierarchical, partition-based, model-based, grid-based, and density-based models. The hierarchical models try to build tree structures, and to generate a hierarchy of groups

according to the criterions of distance, including agglomerative and divisive ways. The well-known hierarchical models are CURE and CHEMELEON [9]. The partition-based models tend to find meaningful partitions in the dataset, and then complete clustering according to specific partitions, e.g., PAM and CLARA [10,11]. The model-based algorithms usually train a model first, based on special mathematical functions on the training dataset, and then complete the clustering, e.g., self-organizing-map (SOM) algorithm [12]. The grid-based models mesh feature space into a finite number of cells, and then cluster data by aggregating similar sub-grids, e.g., CLIQUE [13]. Density-based models group data points according to data's density distribution, and can determine clusters with arbitrary shapes, e.g., typical DBSCAN and OPTICS [14,15]. However, none of these algorithms can be ranked as the most effective one, since all these models have advantages at specific applications. However, orienting to nonlinearly-inseparable data, e.g., clustering spatial data, most of these algorithms have limitations. To address this problem, an enhanced clustering algorithm is generally required.

On the other hand, the computation cost is also a critical issue in clustering, especially in big data analysis. As we know, one disadvantage of traditional clustering algorithms is the high computational complexity [16], which makes them limited in dealing with datasets with a very large size. However, real-world data sources not only have large quantities, but also generate new data continuously as a system operation. Due to the unsupervised character of clustering, some clustering models have to run the whole process once again to determine new data's belonging. In actual engineering applications, it is not practical due to the limitations on RAM and time consumption [17]. An available way is the usage of the model-based methods which can apply the trained model on testing data. In this way, the model-based algorithms can be feasible in online clustering, e.g., utilizing the trained mathematical models to realize new data clustering online [18]. However, to improve the efficiency of clustering, large data are still a problem, e.g., how to accelerate the online clustering process, and how to reduce the total computing overhead.

To address the issues mentioned above in online spatial data clustering, this paper proposes an advanced approach. This method is an extension of the DBSCAN algorithm combined with granular models. Referring to the idea of model-based methods, the proposed method consists of three layers. First, data are clustered by the DBSCAN algorithm in the original data space. Then, each DBSCAN cluster is described by information granules (IG) via granular computing (GrC) [19]. These granules have the structure information from DBSCAN, and are used as structural descriptors. Second, these structural granules are mapped into a hidden layer, which forms rules for determining the data's cluster belonging. In this medium space, the reconstruction metric and justifiable granulating are applied in the fuzzy rule formation. Finally, a rule-based model combining structural granules in input space and rules in medium space is constructed in the output space, and is used to guide the final online clustering. The major contributions of this proposed method are presented as the following three points:

(1) The superiority of using DBSAN algorithms on the nonlinearly-inseparable data: DBSCAN is one of the typical density-based algorithms which group data points of dense regions, and separate regions of low density [14]. It has advantages at solving nonlinear inseparable issues by discovering clusters with arbitrary shapes, namely applicable on spatial data clustering. The second advantage of DBSCAN compared to other methods is the ignorance of expert knowledge on parameter setting, which is helpful to reduce human interference.

(2) The usage of granular models in data description and rule-based modeling: according to human beings' knowledge, GrC divides an object into information granules, and builds granular models for description [20]. By making use of granular models, the shaped clusters in DBSCAN could be broken up into granules which slip the shape's influence. Moreover, granules could reconstruct a cluster with arbitrary shapes. With the information carried in each granule, it is easy to satisfy the expression of new data and complete their online clustering.

(3) The proposed method has an advantage in reducing the computation cost in on-line clustering. Compared with other clustering algorithms, DBSCAN has a relatively acceptable computation complexity, and even has good efficiency on large data sets [21,22]. Additionally, by using a limited number of granules and rules to guide the online clustering, the time of new data clustering can be significantly reduced.

The rest of the paper is structured as follows. In Section 2, some related works on DBSCAN variants are reviewed. Section 3 describes the framework of the proposed method. Moreover, necessary data preprocessing and an introduction on DBSCAN are also presented. Section 4 describes the methodology of the proposed method, involving granule construction in the input space, granular rule formation in the medium space, the mapping and rule-based modeling process in the output space, and the online clustering process. Section 5 presents the experiment studies on both synthetic data and a typical spatial dataset. Finally, Section 6 concludes all of the studies in this paper.

## 2. Related Work on DBSCAN Variants

In this research, the proposed method is actually an extension of the DBSCAN algorithm in online clustering. Therefore, some related work about the development of DBSCAN algorithms are reviewed here, mainly on the modification, improvement, and extension of DBSCAN algorithms.

### 2.1. Basic DBSCAN Algorithm

DBSCAN was first proposed in 1996 [14], and is a density-based clustering algorithm. It generates clusters by the connectivity analysis based on density, namely clustering via the distinction between high-density regions and low-density regions. The main idea is based on the neighborhood of a point, which belongs to a cluster, and contains at least a minimum number of data samples. Therefore, it is easy to determine any arbitrary clusters through this idea. Due to these properties, DBSCAN algorithms are usually applied in clustering data with arbitrary shapes, e.g., in fields involving spatial analysis, such as civil engineering, meteorological engineering, spectroscopy engineering, and diagnostics on medical images. In [23], DBSCAN was applied to study the issue of drinking water distribution pipe breakage in American infrastructure systems. In [24], the DBSCAN algorithm was applied to group stationary GPS trace data into driving destination clusters. In meteorological engineering, DBSCAN was utilized in the application of weather conditions [25,26]. In a spectroscopy application, DBSCAN was used for clustering single particle mass spectra at New York City [27]. Moreover, DBSCAN was proved to be practical in many medical applications, e.g., identification of skin lesion(s), clustering in the diabetes mellitus type II data, and the detection of brain diseases by MRI images [28–30]. Moreover, DBSCAN could be utilized in clustering some non-spatial high-dimension databases. For instance, DBSCAN was used in remote sensing to cluster three-dimensional images in [31]. In [32], DBSCAN was used to monitor anomaly states in high-dimension wind turbine data.

### 2.2. Extensions of DBSCAN Algorithms

However, in some real applications, the original DBSCAN algorithm has some drawbacks, e.g., the poor robustness, and the limitations on clustering data with different densities. Therefore, modifications of the DBSCAN algorithm were proposed in the literature, e.g., VDBSCAN, FDBSCAN, Fast DBSCAN, Grid-DBSCAN, and so on [33,34]. Here, two commonly used variants (Fuzzy-DBSCAN and Grid-DBSCAN) are reviewed.

Fuzzy DBSCAN algorithms were proposed to improve the robustness. In original DBSCAN algorithms, a data point is determined to belong or not belong to a cluster. However, in some special cases where a cluster is adjacent to other clusters, it means that the border points of two clusters are close, or that a border point belongs to several different clusters at the same time. Therefore, the membership of fuzzy theory is usually introduced to deal with this kind of fuzzy concept. The fuzzy DBSCAN algorithms were simply called FN-DBSCAN [35], and they were commonly developed into three types: fuzzy core, fuzzy

boundaries, and a combination of the previous two. The first type of extension model was based on fuzzy core, and it actually utilized fuzzy membership functions to define the density function and redefined core points [36]. It proposed a soft constraint which specified the approximative number of points in the neighborhood of core points. Then, these core points were grouped to fuzzy clusters. Contrary to standard DBSCAN algorithms, which are sensitive to the number of neighbor data points, FN-DBSCAN algorithms are sensitive to the density of neighbor data points. The second type of extension models applied the fuzzy theory to generate the fuzzy boundary [35]. These algorithms defined a membership function on the distance as a soft constraint. This definition made it possible to generate fuzzy clusters with approximate reachability properties and overlapping boundaries. The third type of models combine the previous two fuzziness, namely approximately dense fuzzy cores and reachable fuzzy boundaries [37]. All of these models allow to obtain a much more flexible extension of the DBSCAN algorithm, and to improve the scalability of clustering data points. However, FN-DBSCAN models cost more computation time than standard DBSCAN algorithms due to the fuzzification process. Their results, the same as DBSCAN algorithms, are also not numerical, limiting the extension on online testing for new data points.

Grid-DBSCAN was proposed to deal with the issue that the original DBSCAN algorithms are not effective on grouping clusters with different densities. Its main idea is combining the idea of density-based and grid-based clustering algorithms. Therefore, it was also regarded as a representative hybrid algorithm [38]. It contains two stages to realize clustering of different density groups. At the first stage, it utilizes the grid-based idea to divide feature space into appropriate grids, and the density is similar in each grid. In this way, it forms a multiresolution grid data structure. Then, at the second stage, it aggregates grids with the same densities into groups, and those groups with appropriate density parameters are identified as the final clusters. This is actually implemented by the idea of DBSCAN algorithms. Since the computation time of Grid-DBSCAN relies on the amount of grid cells instead of that of data points, it not only has the ability of clustering different density issues, but also a better computational complexity.

### 2.3. Incremental DBSCAN Variants

Another problem in traditional DBSCAN clustering is the expansibility in online clustering. In today's big data era, the size of the historical dataset is large, so clustering on the whole dataset is a hard task. When new data are generated and added into the studied dataset continually, it is impossible to collect all data points before clustering. For this issue, a good solution is to divide the clustering process into a training part and testing part which aims at clustering new data online. However, since the results of most DBSCAN algorithms are non-parametric, DBSCAN is not applicable in the modeling process. For example, when new data comes, traditional DBSCAN algorithms have to re-cluster the whole dataset again, which certainly decreases its efficiency and wastes a large amount of computation time. Therefore, some extensions of DBSCAN algorithms based on incremental clustering were proposed in the literature, e.g., the incremental DBSCAN algorithm and the incremental G-DBSCAN algorithm [39,40]. These algorithms utilized core points to direct the clustering of new data, and are widely applied to dynamic databases which are updated frequently, e.g., Data Warehouse, anomaly program behaviors detection [41,42]. Compared to a traditional re-clustering approach, the incremental algorithm is verified to be more efficient, since it just needs to cluster new data. The other example was the incremental G-DBSCAN algorithm, which made use of the advantages of the G-DBSCAN model, and realized the clustering of updated data in [40]. Furthermore, considering that most incremental DBSCAN algorithms utilize core points in online clustering, an improved incremental DBSCAN algorithm was proposed based on partition methods in [43], which limit the search space to partitions for obtaining good efficiency.

## 3. Framework and Necessary Preparation

According to the literature review on issues of DBSCAN algorithm variants, an extension algorithm combining DBSCAN with model-based idea is proposed for online clustering in this study, which is implemented with three layers of granule models. The framework of the proposed method is shown in Figure 1.

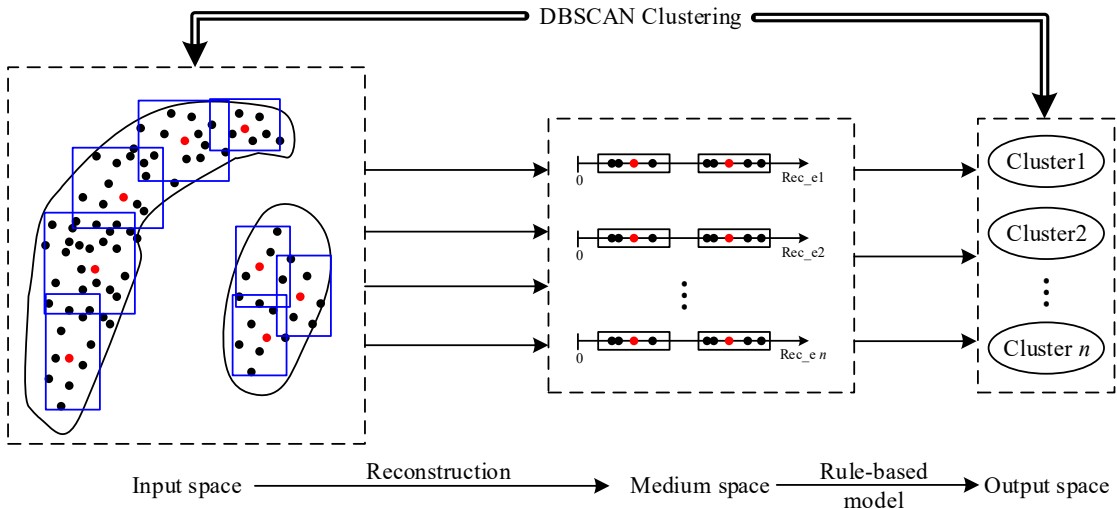

**Figure 1.** Brief framework of the proposed method. red points: prototypes of granules; black points: normal data.

It is seen from Figure 1 that the proposed method is mainly implemented in three spaces: input space, medium space, and output space. First, data are divided into clusters by the traditional DBSCAN clustering algorithm. Labels of these clusters will be regarded to granulate the output space of clustering. Then, the input space is constructed by granule models. Based on the DBSCAN clustering results, the data in the input space are distinguished by arbitrary shapes. Considering that input's dimension is always high, high-dimensional granule models are constructed to describe these structural clusters. Second, a medium space to connect input and output space is established. The medium space could be established by the value domain of an indicator or a function. Since the established medium space is usually one dimension, it is divided into granules expressed by intervals. Each granule in medium space forms rules of determining the mapping from input to output. Finally, according to the model-based idea, cluster labels of new data in testing or online clustering are determined via rule-based modeling in three-layer granule models, including identifying granules in input space, which corresponds to intervals in medium space, and determining cluster labels based on rules in output space.

### 3.1. Data Preprocessing

Before the modeling process, some issues are required to be stated in advance, e.g., data preprocessing, which is a must in data mining and data analysis. There are several necessary data preprocesses, e.g., complement of missing data, correction of abnormal data, de-noising, and normalization. Considering that the raw data in the real world often has different units and magnitudes in clustering, normalization processing is emphasized here. Many normalization methods were proposed in the literature [44], e.g., linear function transformation, logarithmic function transformation, cotangent function transformation, and so on. The max–min function transformation is used the most, and is good at limiting the inputs into a space of $[0,1]^n$. Therefore, it is applied for clustering in this paper, and its formula is defined in (1):

$$y = \frac{x - x_{\min}}{x_{\max} - x_{\min}} \tag{1}$$

where *x* is the original data, and *y* is the value after normalization; $x_{max}$ and $x_{min}$ are the maximum and minimum values. Through Equation (1), the dataset {*y*} will be friendlier in clustering.

### 3.2. Implementation of DBSCAN Algorithm

Based on description of the basic DBSCAN algorithm, a density function is required to realize DBSCAN quantification, which can be denoted as the number of points within a given neighborhood. According to this idea, three types of points via this density function can be denoted, and are shown in Figure 2, namely core points (A), boarder points (B and C), and noise points (N). Assuming a threshold, *thr*, if the density value is larger than *thr*, the point is a core point. On the contrary, the point is a boarder point when it is in a core point's neighborhood, otherwise it is a noise point. Generally, both the core points and their neighbor boarder points are clustered into a cluster, but noise points are not.

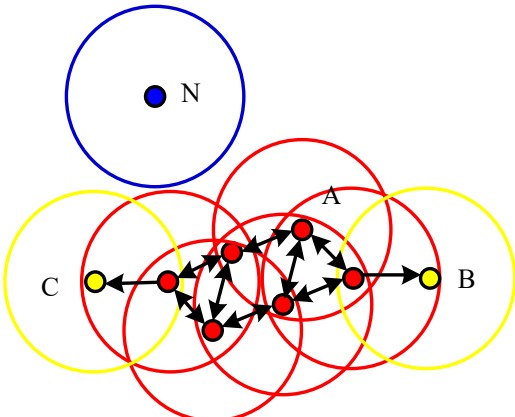

**Figure 2.** Diagram of the basic DBSCAN clustering.

In [14], two parameters related to DBSCAN clustering were defined, the neighborhood size (*s*) and the density threshold (*thr*). Then, given a test point, *p*, its density function can be expressed as $N_s(p)$ in (2):

$$N_s(p) = \text{Num}(\{q \in \text{T} \mid dis(p,q) \leq s\}) \tag{2}$$

where *q* is another arbitrary point in *T*; *dis(p,q)* is the distance between these two points (*p* and *q)*; and *Num*(\*) is denoted as the function to count the number of points. Then, all points can be categorized and grouped to clusters.

## 4. Methodology of the Proposed Three-Layer Granule Models

Based on the above description, the DBSCAN algorithm is utilized to cluster training data firstly, and its results determine how data aggregates into clusters in the input space, and how many clusters will be in the output space. Then, the main task in modeling is to construct information granules in the input and medium spaces.

### 4.1. Granulating in the Input Space

DBSCAN can generate clusters that have arbitrary shapes, and its output is non-parametric. This causes difficulty on its extension for online clustering, unlike *k*-means or fuzzy *c*-means (FCM) [45], which can utilize cluster centers for online clustering directly. In the literature, several incremental DBSCAN algorithms were proposed to utilize core points for clustering updated data. However, the number of core points in big data is still huge. Therefore, this paper proposed to granulate core points, and to use prototypes of granules to direct online clustering. Contrary to incremental DBSCAN algorithms based on grids and partitions, granules in this study have noting to do with shapes of original clusters, but they could be used to reconstruct the shaped clusters. On the other hand, since

an information granule is a set of similar points, the computation cost will be reduced by only considering the prototypes of granules in GrC.

### 4.1.1. Construction of Information Granules (IGs)

The process of granulating is described as follows. Assuming a cluster resulting from DBSCAN, several data points are selected from the cluster as its descriptors firstly. If the cluster has $N$ data points, the selected data points are formed as a set of prototypes, $Z$, expressed in (3):

$$Z = \{z_1, z_2, \cdots, z_c\} \tag{3}$$

where $z_i$ represents the prototype of the $i$th cluster; $c$ is the number of selected data points, and usually set as a relatively small value for the consideration of efficiency.

Then, based on the selected prototypes, data points can be grouped into different IGs, which are constructed centering on those prototypes. To realize IG construction, the membership of data is a useful tool for expressing data's belonging to IGs, e.g., the fuzzy membership definition in (4). Assuming a data point, $x_k \in X_h$, where $X_h$ is the $h$th cluster in DBSCAN clustering results, its membership function to $IG_i$ is defined as (4):

$$u_i(k) = \frac{1}{\sum\limits_{j=1}^{c} \left( \frac{\|x_k - z_i\|}{\|x_k - z_j\|} \right)^{\frac{2}{m-1}}} \tag{4}$$

where $u_i(k)$ is the value of membership expressing how close $x_k$ is to the $i$th granule; $m$ is the fuzzification parameter, which is required to be larger than 1. Considering the prototypes are directly regarded as cores of granules, $m = 2$ is generally determined [20].

Considering that there exists a case that a point is equal to the prototype, namely $x_k = z_r$, where $z_r$ is a prototype, the function in (4) will not be able to compute because of $\|x_k - z_r\| = 0$. To address this problem, a modification on (4) is defined as below:

$$u_i(k) = \begin{cases} 0, & x_k = z_r, z_r = z_i \\ 1 / \sum\limits_{j=1}^{c} \left( \frac{\|x_k - z_i\|}{\|x_k - z_j\|} \right)^{2/(m-1)}, & x_k \neq z_r \\ 1, & x_k = z_r, \ z_r \neq z_i \end{cases} \tag{5}$$

The new function degenerates to distinct membership as $x_k = z_r$. When $z_r = z_i$, the membership, $u_i(k)$, can be directly set as 1. When $z_r \neq z_i$, implying this point overlaps other prototype points, the membership can be set as $u_i(k) = 0$ and $u_r(k) = 1$. Then, after the re-calculation on the membership, data points can be partitioned into different groups centering on prototypes, and each group is regarded as a fuzzy granule.

### 4.1.2. Determination of the Granule Size

Generally, IGs are designed as hyperrectangles in the high-dimensional input space [16]. Expanding around a selected prototype, $z_i$, the corresponding granule in the input space could be built as a hyperrectangle, $B_{hi} = [l_{hi}, u_{hi}]$. Considering that all data are normalized, the constraint of boundary is also satisfied, namely $l, u \in [0,1]^n$.

Data points are partitioned into different granules by membership matrix, but the obtained information granules are fuzzy. Therefore, the performance of granule construction will be directly affected by the size of the granules. In general, two indicators can be proposed to construct and evaluate IG's performance: coverage and specificity [22].

Coverage is an indicator representing the amount of data points covered in an IG, so it can be simply defined as below:

$$Cov = \frac{1}{N} \sum\limits_{k=1}^{N} incl(t_k \in Y_k) \tag{6}$$

where $Y_k$ represents the cluster; $t_k$ is the output; and $incl(*)$ is a Boolean function. When $t_k \in Y_k$, the value of $incl(*)$ is 1.

Specificity is the other indicator. Generally, it requires a high value when the granule size is small. Therefore, we can define a simple function as below:

$$Spe = \frac{1}{N}\sum_{k=1}^{N} \exp(-\alpha * size(Y_k)) \tag{7}$$

where $scale(Y_k)$ is the IG size; $exp(*)$ is a selected function expressing the negative correlation between IG size and specificity, and $\alpha$ is a slope control factor. For example, information granules are designed as hyperrectangles in this paper, and the selected points are prototypes. Thus, the size of the information granules could be decided by a vector, $S$, as shown in (8):

$$S = [\varepsilon_1^+, \varepsilon_1^-, \varepsilon_2^+, \varepsilon_2^-, \cdots, \varepsilon_n^+, \varepsilon_n^-] \tag{8}$$

where $\varepsilon_i^+$ and $\varepsilon_i^-$ are the distances of $z$ to the upper bounder, $u_i$, and lower bounder, $l_i$, respectively.

For example, a two-dimensional granule is shown in Figure 3a. A good information granule usually requires high coverage and high specificity. However, there exists a conflict to reach the highest values of both the indicators, as shown in Figure 3b.

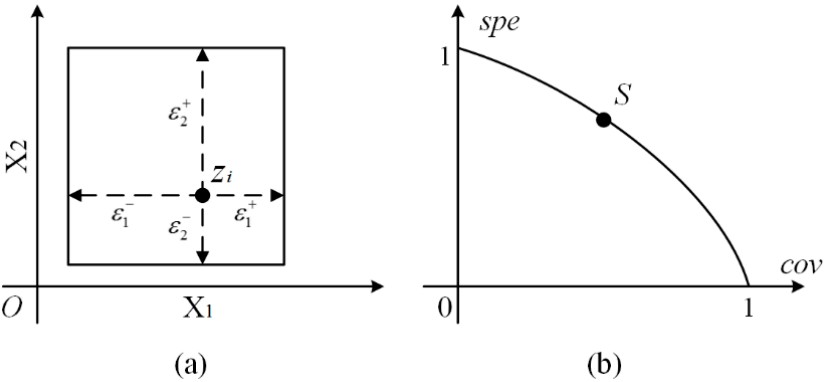

(a)                                                        (b)

**Figure 3.** (**a**) A rectangle information granule is constructed by the prototype, $z_i$, and its size is determined by four elements $[\varepsilon_1^+, \varepsilon_1^-, \varepsilon_2^+, \varepsilon_2^-]$ in vector, $S$; (**b**) the conflict relation of specificity and coverage when determining the size of granules.

In order to address this multi-objective problem, an alternative design is to consider a product of these two indicators, defined in (9):

$$V(S) = Cov * Spe \tag{9}$$

where $V(S)$ is defined as a new indicator determining the granule size, $S$.

### 4.2. Granulating in the Medium Space
#### 4.2.1. Construction of the Medium Space

As seen in the description in Figure 1, the medium space is a hidden layer which has no clear definition. Its purpose is to build the connection between input space and output space, and to realize the model-based clustering. Therefore, we could choose some mathematical functions or performance indicators to construct the medium space. The granules of input space not only wipe out the shape influence by dividing original clusters, but can also be used to reconstruct the original clusters with arbitrary shapes. It is seen that

reconstruction is a perfect bridge connecting the two spaces. Therefore, a reconstruction indicator is proposed for evaluating the performance quantitatively here, as defined in (10):

$$Rec = \frac{1}{card(X)} \sum_{x \in X} \|x - \hat{x}\|^2 \tag{10}$$

where *Rec* is the reconstruction error; *card(X)* is the function counting data amount in the dataset, *X*. This indicator requires that a good reconstruction should have a small *Rec* value.

It is seen from (10) that, if the membership matrix and prototypes are given, the reconstructed data point is expressed in (11):

$$\hat{x}_k = \left( \sum_{i=1}^{c} u_i^m(x_k) \cdot z_i \right) / \left( \sum_{i=1}^{c} u_i^m(x_k) \right) \tag{11}$$

where $\hat{x}_k$ is the reconstructed data of the original data, $x_k$. Equation (11) is to reconstruct a data point by the weighted average of the prototypes, and the weights are calculated as $u_i^m(x_k) / \sum_{i=1}^{c} u_i^m(x_k)$. According to this concept, the reconstructed data will be different when using descriptors of different clusters. If the original data belong to a cluster, the reconstruction error calculated by granules of the corresponding cluster would be small. Therefore, the medium space is constructed by the reconstruction error of all output clusters in this paper.

### 4.2.2. Justifiable Granulating

Since the medium space constructed by reconstruction error is one dimension, the granules in this hidden layer are intervals. To construct suitable granule models, the size of interval granules is required to be optimal, which is actually decided by the bound of intervals according to (8). In [46], a method of constructing interval information granules was proposed, called justifiable granulating. It also utilized Equations (6), (7), and (9) to find the optimal granule size. Contrary to high-dimensional granules, no prototypes are chosen in interval granules. Therefore, the median of data is chosen as the numerical representation in [46], and regarded as prototypes of interval granules. Then, the size vectors, $S = [\varepsilon^+, \varepsilon^-]$ (lower and upper bound, respectively), are calculated by Equations (6), (7), and (9). Since two parameters, $\varepsilon^+, \varepsilon^-$, need to be decided in $S$, two objective functions are given out in Equations (12) and (13):

$$V(\varepsilon^+) = f_1(card\{x_k \in [y_m, y_m + \varepsilon^+]\}) * f_2(|\varepsilon^+|) \tag{12}$$

$$V(\varepsilon^-) = f_1(card\{x_k \in [y_m - \varepsilon^-, y_m]\}) * f_2(|\varepsilon^-|) \tag{13}$$

where $y_m$ is the median value; $f_1$ and $f_2$ represent the coverage function (6) and specificity function (7), respectively. In interval granules, the function, *incl*(*), is expressed as the *card*{$Y_k$}, and the function *size*($Y_k$) is expressed as the length of $Y_k$, as $size(Y_k) = |y_k^+ - y_k^-|$. The optimal lower and upper bound are obtained by maximizing (12) and (13), respectively.

### 4.3. Online Clustering via Rule-Based Models

Based on the description in Sections 4.1 and 4.2 the connection between the input and medium spaces is established by their information granules. On the other hand, the step of testing or online clustering is to determine the cluster labels according to the reconstruction errors, and it could be regarded as a rule-based process from the medium space to the output space. Generally, a fuzzy rule-based system is constructed by linguistic rules described in (14):

$$IF\ premise\ (antecedent)\ THEN\ conclusion\ (consequent) \tag{14}$$

where the inference is described as: if the fact (*premise*) is given out, then the *conclusion* can be determined. Therefore, Equation (14) is also called as IF-THEN rule-based form.

According to the above concept, rules could be established connecting the medium space and the output space. The conclusion is certainly the cluster label in the clustering problem. The premise in this paper is decided in the medium space. Taking each data point as the study object independently, it is determined to belong to a cluster in the output space or not. Therefore, for a given cluster, any data point is distinguished into belonging and not-belonging, and these naturally correspond to two rules. To express these rules concretely, interval granules constructed by reconstruction errors in the medium space are considered. The final rules of a rule-based model for clustering are formed as below:

$$R_i : \text{ IF } Rec(x_k) \text{ is in } IG_{i,1}, \text{ THEN } x_k \text{ is in } C_i;$$
$$\text{IF } Rec(x_k) \text{ is in } IG_{i,0}, \text{ THEN } x_k \text{ is not in } C_i;$$
(15)

where $C_i$ represents the $i$th cluster; $R_i$ represents the rule library of cluster $C_i$; $Rec(x_k)$ is the reconstruction error of $x_k$ based on $IG$s in cluster $C_i$; $IG_{i,1}$ is the interval granule consisting of all points in cluster $C_i$; and $IG_{i,0}$ is the interval granule involving all points outside of cluster $C_i$. If the reconstruction error locates in the interval granule of belonging to a cluster, then the final cluster label is determined. The flow sketch of online clustering is shown in Figure 4.

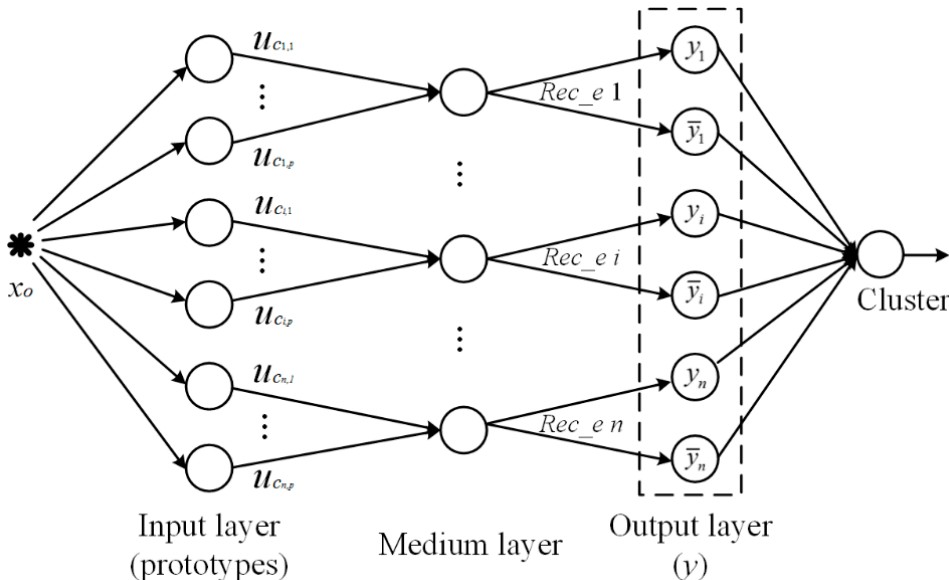

**Figure 4.** Flow sketch of online clustering by the proposed rule-based model.

According to Figure 4, the process of online clustering is described. First, for new testing data, its memberships to granules in the input space are calculated. Then, its reconstruction error based on different clusters is calculated in the medium layer. Finally, according to the established rules library in (15), the data point is determined to the corresponding cluster, and its cluster label is the output. Since the whole process is based on memberships and fuzzy rules, the online clustering approach is also called the fuzzy rule-based model.

## 5. Experiment and Discussion

### 5.1. Synthetic Data

To study the performance of the proposed method, a synthetic dataset could be given for the primary experiments. For example, a dataset having two two-dimensional clusters is defined in (16):

$$Cluster1 : \begin{cases} x_1 = \cos(\theta) + \varepsilon; \\ x_2 = \sin(\theta) + \varepsilon; \end{cases}$$
$$Cluster2 : \begin{cases} x_1 = 1 - \cos(\theta) + \varepsilon; \\ x_2 = 0.5 - \sin(\theta) + \varepsilon; \end{cases} \tag{16}$$

where $x_1$ and $x_2$ are two dimensions; $\varepsilon$ is the Gaussian noise; $\theta$ is an angle variable. One cluster is the rotation and translation of the other cluster. Here, 2000 training data points are generated by (16), 1000 for Cluster 1 and 1000 for Cluster 2. Then, the dataset is normalized by (1), and is plotted in Figure 5.

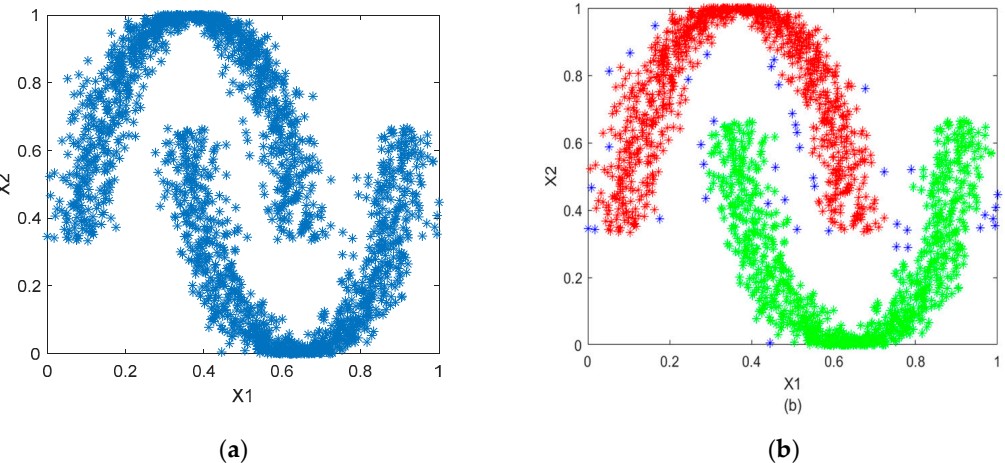

(a)                                                    (b)

**Figure 5.** Distribution of data points in toy dataset. (**a**) Original data distribution; (**b**) DBSCAN clustering results (two clusters in red and green points, noise in blue points).

In Figure 5, the distribution of data points is shown: Figure 5a shows the original data and Figure 5b shows the results via DBSCAN clustering. It is seen that there are obviously two clusters which are not linearly separable. Traditional clustering algorithms (e.g., *k*-means, FCM) are not applicable in this case, related to the clustering issue on spatial datasets, but DBSCAN algorithm has superiority. To obtain the optimal parameters in DBSCAN clustering, the *k*-dist graph method [47] is generally used, e.g., parameters (k = 5, s = 0.0273) are applied for DBSCAN clustering in Figure 5b.

### 5.1.1. Optimal Prototypes and Granules Construction of Input Space

After DBSCAN clustering, clusters with arbitrary shapes are obtained. In order to make use of these cluster information in online clustering, some representative points are selected to describe these clusters, called prototypes in the above context. At the beginning, we could choose these prototypes randomly, and express them as (3). Then, these prototypes are adjusted to obtain the strongest representativeness. Finally, an information granule centering on this prototype is constructed. The concrete algorithms are described as follows.

Step 1: initiating prototypes and IGs. $c$ prototypes in each cluster are chosen randomly, denoted as $\{z_i \mid I = 1, 2, \cdots, c\}$. Then, IGs are constructed according to the description in Section 4.1. To initialize the IGs' size, the upper and lower bounds are calculated in a dimension-by-dimension way, seeing the granule construction in two-dimensional space, as seen in Figure 6. Firstly, the initialized prototypes and whole data points are projected at each one-dimension axis, and sorted in ascending order, e.g., $\{\min(X_1), z_{1,1}, z_{2,1}, z_{3,1}, z_{4,1}, \max(X_1)\}$ at axis $X_1$. Then, the two closet prototypes are selected to compute their upper (or lower) bound through (17):

$$u_{i,k} = l_{i+1,k} = \frac{z_{i,k} + z_{i+1,k}}{2} \tag{17}$$

where $z_{i,k}$ and $z_{j,k}$ are the values of two prototypes ($z_i$ and $z_{i+1}$) at axis $X_k$. This process is repeated to determine the bounds of all IGs. When the prototype, $z_i$, is closet to 0/1, its bounds can be determined directly as $l_{i,k} = \min(X_k)$ or $u_{i,k} = \max(X_k)$, e.g., $l_{1,1}$ and $u_{4,1}$ in Figure 6.

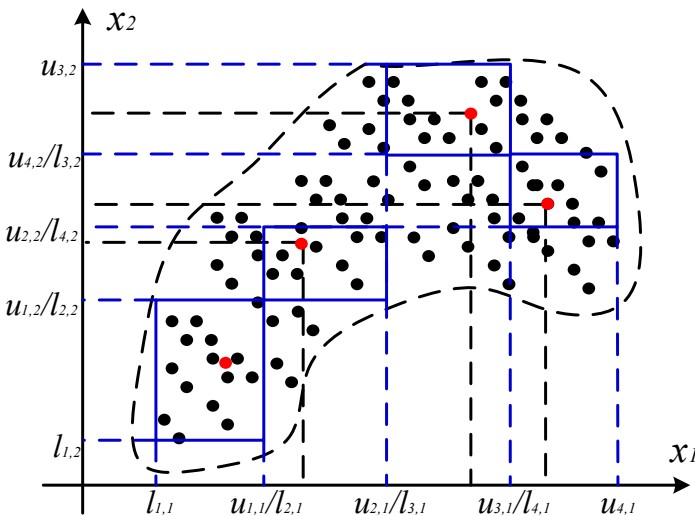

**Figure 6.** Initialization of granule size in two-dimension space. red points: prototypes of granules; black points: normal data.

Step 2: updating prototypes. In Figure 6, it is seen that the initialized prototypes (red points) are not always the centroids of IGs. In order to improve the prototypes' representativeness, original prototypes can be updated after granule construction, as below:

$$z_i' = \left[ \frac{u_{i,1}+l_{i,1}}{2}, \quad \frac{u_{i,2}+l_{i,2}}{2}, \quad \dots, \quad \frac{u_{i,n}+l_{i,n}}{2} \right] \tag{18}$$

where $z_i'$ is the new prototype which is the centroid of the corresponding granule. Moreover, considering the randomness in the initial selection, the prototypes are not distributed uniformly, so the coverage and specificity performance may be affected. For example, when two prototypes close to each other, their granules are easily overlapped. Taking these issues into account, prototypes need to be updated further. Here, to evaluate the suitability of the updated prototypes, a metric is defined in (19) which requires the distance of all data points to their belonging prototypes as the minimum:

$$\min Q = \sum_{i=1}^{c} \sum_{x \in G_i} \|x - z_i\|^2 \tag{19}$$

where $Q$ is the objective function for updating prototypes; $G_i$ is the $i$th information granule based on prototype $z_i$; $c$ is the number of granules; $\|x - z_i\|^2$ is the Euclidean distance of point x to its corresponding prototype. When this objective function converges to a given threshold, the updated prototypes are regarded as optimal.

Step 3: adjusting the size of IGs. By repeating Step 1 and Step 2, the optimal prototypes are determined, and IGs can be constructed. However, it is seen from Figure 6 that these primary granules can not achieve a good coverage, so it is necessary to adjust the size of granules based on Equations (6)–(8). Though there are $2 \times n$ parameters in (8), it will cost a lot of computation overhead in the adjustment process. Considering the prototypes are updated as the centroids of granules in Step 2, we could consider only one parameter here to adjust the granule size, namely the increment parameter ($\varepsilon$) to the boundary. Then, the optimal granule sizes are updated based on (9).

Figure 7 shows the results of granule computing on the toy data; the IG construction results and the initial and optimal IGs are plotted in Figure 7a,b, respectively. Quantitatively,

Table 1 presents the numerical results of the optimal prototypes and the size of the IGs finally. From these results, some conclusions can be obtained. First, the selected prototypes can well reflect the distribution of data, and the corresponding IGs can well describe data structures. This verifies the feasibility of using GrC in spatial data mining. Second, comparing IGs in Figure 7a,b, it is seen that IGs after optimization can have higher specificity, and data coverage is also guaranteed, which implies the proposed GrC steps and optimization methods are effective.

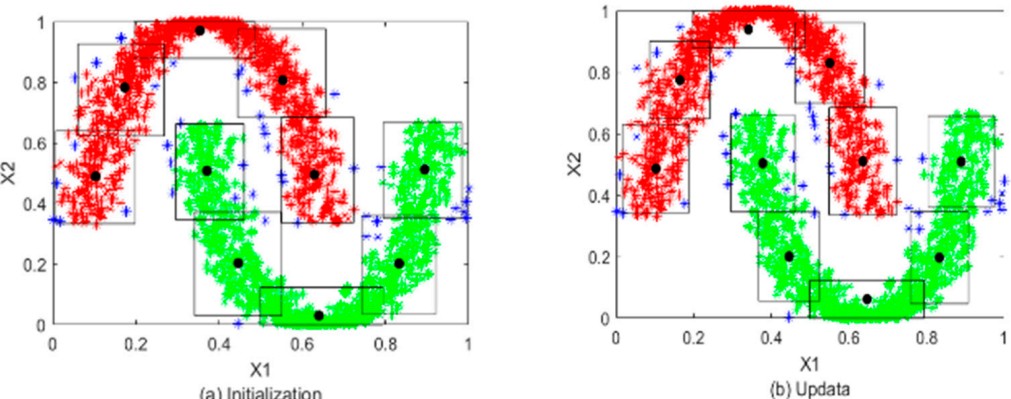

**Figure 7.** Results of granular computing on the toy data. (**a**) Initial IG; (**b**) optimal IG. Where, two clusters are represented by red and green points, noise is represented by blue points.

**Table 1.** Prototypes and information granules of two clusters ($c$ = 5).

|  | $z_1$ | $z_2$ | $z_3$ |
|---|---|---|---|
| Cluster1 | (0.5507, 0.8299) | (0.1020, 0.4865) | (0.6361, 0.5103) |
| Cluster2 | (0.4456, 0.2005) | (0.8895, 0.5088) | (0.8331, 0.1971) |
|  | G1 | G2 | G3 |
| Cluster1 | [0.4606, 0.6408] × [0.6993, 0.9605] | [0.0160, 0.1881] × [0.3429, 0.6302] | [0.5487, 0.7236] × [0.3354, 0.6852] |
| Cluster2 | [0.3668, 0.5244] × [0.0557, 0.3452] | [0.8040, 0.9750] × [0.3605, 0.6572] | [0.7584, 0.9078] × [0.0471, 0.3471] |
|  | $z_4$ | $z_5$ |  |
| Cluster1 | (0.3409, 0.9394) | (0.1640, 0.7746) |  |
| Cluster2 | (0.3778, 0.5040) | (0.6467, 0.0618) |  |
|  | G4 | G5 |  |
| Cluster1 | [0.1948, 0.4869] × [0.8788, 1.0000] | [0.0861, 0.2420] × [0.6489, 0.9004] |  |
| Cluster2 | [0.2959, 0.4598] × [0.3455, 0.6625] | [0.4993, 0.7941] × [0, 0.1237] |  |

### 5.1.2. Construction on Interval Granules in Medium Space

According to the description in Section 4.2, the medium space is constructed by reconstruction errors. Therefore, the memberships of data points to all prototypes are computed firstly. Considering the purpose of medium space is to construct fuzzy rules for online clustering, the reconstruction error is calculated focusing on each cluster. For example, assuming a data point, $x$, its membership matrix is calculated as U = {$u_i$}. To discuss its affiliation to the $k$th cluster, only the memberships relating to granules of Cluster $k$ are considered, as shown in (20):

$$u_k = [u_{(k-1)*c+1}, u_{(k-1)*c+2}, \cdots u_{k*c}]; \tag{20}$$

where $u_k$ represents the membership matrix to granules belonging to Cluster $k$. Then, reconstruction errors of the dataset based on $u_k$ are calculated according to Equations (10)

and (11). Based on two moon clusters in the toy data, reconstruction errors of each cluster are calculated independently, and are shown in Figures 8 and 9.

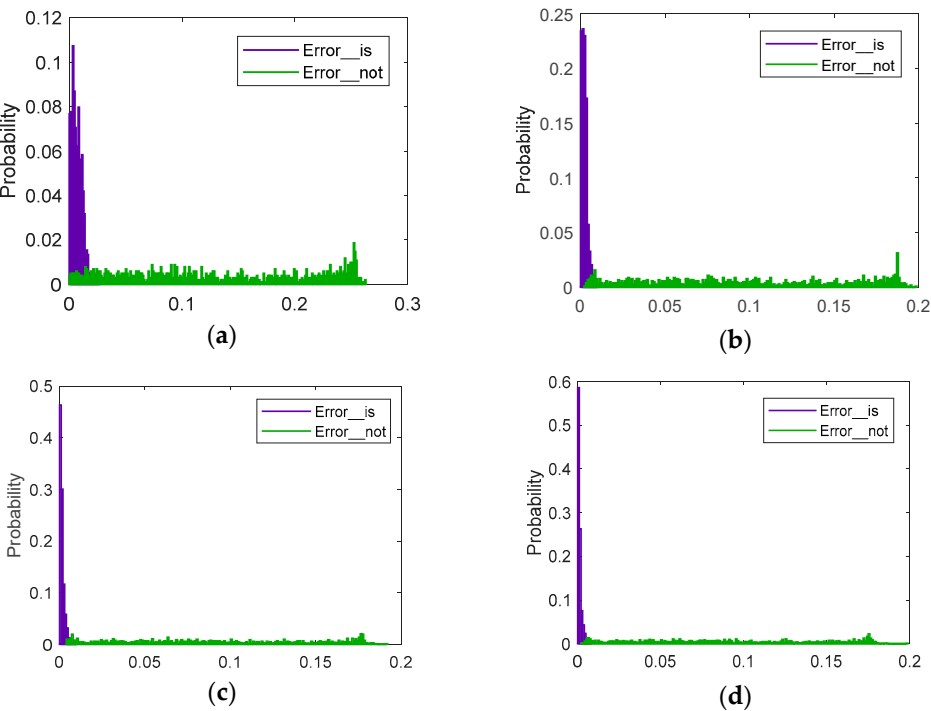

**Figure 8.** Statistics of reconstruction errors in Cluster 1. (**a**) $c = 5$; (**b**) $c = 10$; (**c**) $c = 15$; (**d**) $c = 20$.

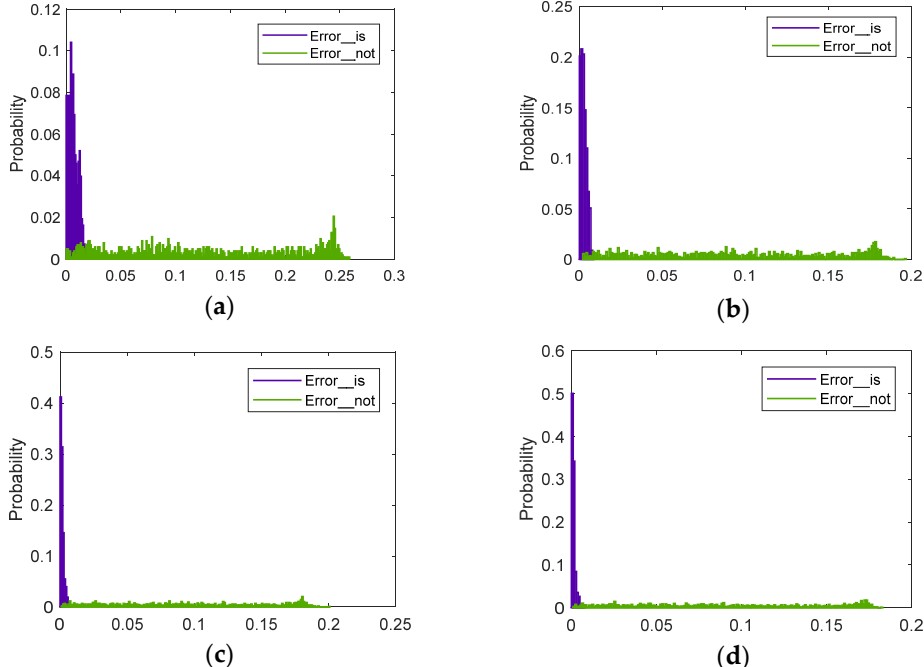

**Figure 9.** Statistics of reconstruction errors in Cluster 2. (**a**) $c = 5$; (**b**) $c = 10$; (**c**) $c = 15$; (**d**) $c = 20$.

In Figures 8 and 9, *Error_is* and *Error_not* represent the reconstruction errors of data points belonging/not-belonging to a given cluster, respectively. Four sub-figures show the statistics of reconstruction errors with different numbers of IGs ($c = 5, 10, 15, 20$), respectively. It is seen that the distinguishability in the data description is enhanced when the number of IGs increases, which also implies that it is feasible to make use of the reconstruction errors (*Error_is* and *Error_not*) to form rules for clustering.

First, according to the distribution of *Error_is* and *Error_not*, we could construct the interval granules in the medium space as Section 4. It is seen from Figures 8 and 9 that reconstruction errors of points within a cluster (*Error_is*) are mostly close to zero. Therefore, the lower bound of the interval granule is set as 0, and the upper bound is calculated by justifiable granulating methods as Equations (12) and (13). Conversely, it is also seen that the reconstruction errors of data points outside a cluster (*Error_not*) distribute symmetrically. Therefore, the lower bound and upper bound of interval granules can be set symmetrically to the median value, $y_m$. On the other hand, if the statistical distributions of *Error_is* and *Error_not* in Figures 8 and 9 are assumed to fit some probability density functions (PDF), then the objective functions in Equations (12) and (13) for calculating bounds are rewritten by the following continuous forms:

$$\begin{cases} f_1(\varepsilon) = \int\limits_{\overline{x} \to \varepsilon} p(x)dx \\ f_2(\varepsilon) = \exp(-\alpha \cdot \varepsilon) \end{cases} \tag{21}$$

Based on Equations (12), (13), and (21), the bounds of interval granules (*Error_is* and *Error_not*) are calculated. According to the above description, the interval granule (*Error_is*), representing points within the *i*th cluster, is expressed as $B_{i1}$, and the interval granule (*Error_not*) is defined as $B_{i0}$ on the contrary. Their bounds in the medium space can be further expressed in detail, as (22):

$$B_{i1} = [0, V(\varepsilon_1^+)]; \quad B_{i0} = [V(\varepsilon_0^-), V(\varepsilon_0^+)]; \tag{22}$$

where $V(\varepsilon_1^+), V(\varepsilon_0^-), V(\varepsilon_0^+)$ are the optimal bounds calculated via GrC. The final numerical results are presented in Table 2, where $IG_y$ and $IG_n$ present the granular intervals of *Error_is* and *Error_not*.

**Table 2.** Results of granular intervals in the medium space.

|  | *c* = 5 | *c* = 10 |
|---|---|---|
| Cluster1 | $IG_y$: [0, 0.0260]; $IG_n$: [0.0011, 0.2591]; | $IG_y$: [0, 0.0090]; $IG_n$: [0.0034, 0.1914]; |
| Cluster2 | $IG_y$: [0, 0.0270]; $IG_n$: [0.0046, 0.2526]; | $IGy$: [0, 0.0080]; $IG_n$: [0.0055, 0.1875]; |
|  | *c* = 15 | *c* = 20 |
| Cluster1 | $IG_y$: [0, 0.0090]; $IG_n$: [0.0044, 0.1804]; | $IG_y$: [0, 0.0070]; $IG_n$: [0.0036, 0.1796]; |
| Cluster2 | $IG_y$: [0, 0.0070]; $IG_n$: [0.0047, 0.1887]; | $IG_y$: [0, 0.0070]; $IG_n$: [0.0029, 0.1789]; |

Second, based on these constructed interval granules, we could build two fuzzy rules for each cluster, e.g., one rule for determining a data point belonging to a cluster, and one rule for not-belonging, as presented below:

$Rules\ for\ the\ i_{th}\ cluster\ C_i:$
$$\begin{cases} R_i: \text{ IF } Rec(\boldsymbol{x}_k) \text{ is in } B_{i1}, \text{ THEN } \boldsymbol{x}_k \text{ belongs to } C_i; \\ \overline{R}_i: \text{ IF } Rec(\boldsymbol{x}_k) \text{ is in } B_{i0}, \text{ THEN } \boldsymbol{x}_k \text{ doesn't belong to } C_i; \end{cases}$$

where $R_i$ is a rule used to determine if $\boldsymbol{x}_k$ belongs to cluster $C_i$; $\overline{R}_i$ is the contrary rule used to determine that $\boldsymbol{x}_k$ does not belong to cluster $C_i$. These two rules are both fuzzy rules. With the help of these rules, when new data are obtained in online testing, clustering on these data can be quickly completed with less cost. Moreover, due to the fuzzy characteristics, the final decision can allow an extent of uncertainty when facing some confused data requiring manual checking, instead of making a determined decision blindly.

5.1.3. Online Testing and Evaluation

According to the description in Section 4.3, the online testing is actually a rule-based clustering process. It is to determine the belonging clusters of testing data based on rules. Since the proposed approach considers DBSCAN and GrC in rule formation, it of course inherits the ability of density-based methods on distinguishing arbitrarily-shaped clusters. However, the rule-based clustering is superior to density-based clustering on reducing the cost of online clustering. For example, the crispy DBSCAN algorithm has to test all core points when clustering new incremental data, but the proposed approach can make decisions by just the constructed rules.

Based on Equation (16), 1000 new data points are generated as the testing dataset, which contains 500 points in Cluster 1 and 500 points in Cluster 2, respectively. The concrete online clustering of the proposed method is described in Figure 4. It is known that the final result could be calculated by rules of both $R_i$ and $\overline{R}_i$, and it can be fuzzy or determined according to the application requirement. In this paper, we just consider the rule, $R_i$, to obtain a certain cluster label. To evaluate the performance of clustering the new incremental dataset quantitatively, the accuracy metric can be given out in (23):

$$Acc = \frac{1}{N}\sum_{i=1}^{nc} TP_i \times 100\% \tag{23}$$

where *Acc* is the accuracy; *nc* is the amount of clusters; *N* is the total amount of samples for testing; and *TP* represents true positive event via the confusion matrix [48], i.e., it calculates the amount of correctly-clustered data points.

Table 3 shows the performance of the proposed approach in online clustering, including the accuracy and computation time. Because different numbers of granules have different influences on the specificity of rules in medium space (as shown in Figures 8 and 9), this subsequently affects the final online clustering performance. Therefore, four different parameters are discussed here: $c = 5, 10, 15, 20$. From the above results, it is found that as the number of granules increase, the accuracy of online clustering improves, but the computation cost will also increase. Therefore, there should be a trade-off for the final selection in practical applications.

**Table 3.** Performance of the proposed approach in online clustering.

|  | $c = 5$ | $c = 10$ | $c = 15$ | $c = 20$ |
|---|---|---|---|---|
| Acc. | 0.8920 | 0.9350 | 0.9400 | 0.9520 |
| Time | 0.0316 | 0.0438 | 0.0908 | 0.0927 |

To further compare the performance of the proposed approach with other DBSCAN extensions on online clustering, as well as other well-known online clustering methods, four methods are selected here, such as crispy incremental DBSCAN (C-DBSCAN), incremental grid-DBSCAN algorithm (G-DBSCAN), online *k*-means, and online weighted *k*-means (W-*k*-means) [49].

Table 4 shows the comparison results of different online clustering methods on the given toy data. The proposed approach is set up with $c = 20$ for the sake of good performance. By comparing with the other two DBSCAN extensions, which also have the same number of cells for online clustering, it is found that the major superiority of the proposed approach is less time consuming, even though it also has a slightly better online clustering performance than other two. In this way, it is not difficult to understand that the proposed method would achieve bigger superiority on online clustering overhead when dealing with a larger size of data in big data analysis. On the other hand, by comparing results of the proposed method with that of two well-known online clustering methods, it is found that the proposed method has no superiority on computation time, since *k*-means and its extensions can use cluster centers to guide online clustering fast, whereas the proposed

method can obviously outperform *k*-means extensions on the accuracy performance. The reason behind this phenomenon is the ability of granules on data structure description, whereas the cluster labels are also denoted according to the distinction on data structure. However, the centers of *k*-means clusters are randomly distributed without considering data structure information, and DBSCAN-based granules can make use of data density and structure description ability to realize good performance on spatial data online clustering.

**Table 4.** Performance comparison on different online clustering methods.

|  | C-DBSCAN | G-DBSCAN | Proposed Approach | Online *k*-Means | Online W-*k*-Means |
|---|---|---|---|---|---|
| Acc. | 0.9450 | 0.9330 | 0.9520 | 0.8947 | 0.9058 |
| Time | 0.2279 | 0.1712 | 0.0927 | 0.0632 | 0.0675 |

*5.2. Spatial Benchmark Data*

Furthermore, to study the generalization ability of the proposed method, especially the superiority of the proposed method on spatial data online clustering, a typical spatial data (t4.8k dataset) is considered here, namely the Chameleon dataset [50]. This dataset contains totally 8000 data points, and two attributes are generally taken for experiments.

Figure 10 shows the distribution of Chameleon data. It is seen from Figure 10a that this dataset has some nested shapes which are not linearly-separable. Therefore, density-based models can be applied for its clustering in the literature [41], e.g., DBSCAN is used in this paper. The results are shown in Figure 10b, where six shaped clusters are obtained. According to the description in Section 3, each cluster is taken for study individually, and the optimal prototypes of IGs are presented in Table 5.

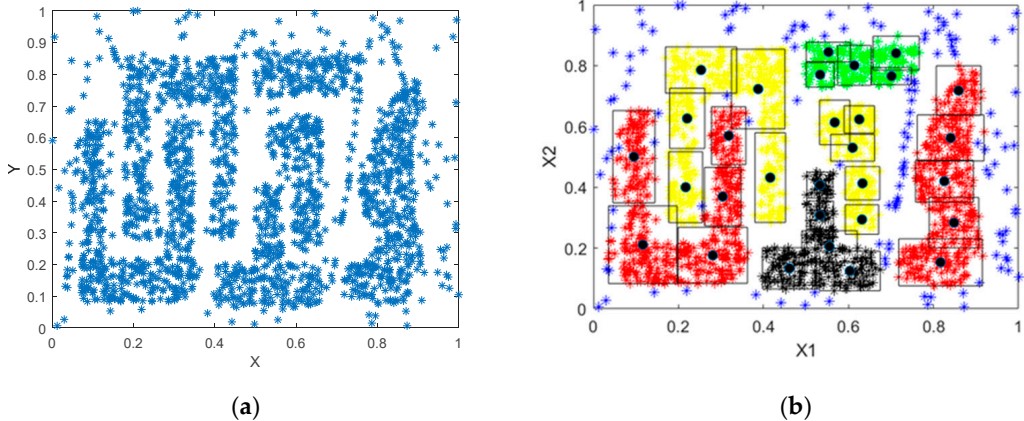

(**a**)                                                                                          (**b**)

**Figure 10.** Distribution of data in Chameleon dataset. (**a**) Original data distribution; (**b**) DBSCAN clustering result.

For the sake of convenience on the presentation and visualization, the results of Table 5 are based on a specific number of granules, i.e., *c* = 5 in each shaped cluster. Then, based on these optimal prototypes and the mentioned procedures of GrC, granules in each structural cluster are constructed, shown in Figure 10b. It is seen that these granules construct the skeleton of data distribution, and effectively describe the data structure of the spatial data. The noise data (anomaly data) outside the normal distribution are effectively excluded, which is also the superiority of the proposed method to other clustering methods (such as *k*-means, FCM). Then, according to granules in the input space, granular fuzzy rules in the medium space can be correspondingly constructed and prepared for online clustering. By utilizing justifiable granulating on the reconstruction error metric, granular intervals for determining if a data point belongs to a given cluster or not are computed. The results are presented below.

**Table 5.** Optimal prototypes in each cluster.

|  | $z_1$ | $z_2$ | $z_3$ | $z_4$ | $z_5$ |
|---|---|---|---|---|---|
| C1 | (0.0953, 0.5005) | (0.3181, 0.5696) | (0.2808, 0.1761) | (0.3042, 0.3694) | (0.1163, 0.2112) |
| C2 | (0.5333, 0.7704) | (0.7013, 0.7656) | (0.712, 0.841) | (0.6139, 0.8009) | (0.553, 0.8449) |
| C3 | (0.5329, 0.3078) | (0.6032, 0.1249) | (0.5328, 0.4072) | (0.4611, 0.1334) | (0.5553, 0.2053) |
| C4 | (0.2533, 0.7857) | (0.217, 0.4002) | (0.4159, 0.4317) | (0.2205, 0.6264) | (0.3876, 0.7231) |
| C5 | (0.6318, 0.294) | (0.6254, 0.6233) | (0.6097, 0.5299) | (0.5676, 0.6134) | (0.6329, 0.4129) |
| C6 | (0.8482, 0.2836) | (0.8402, 0.5621) | (0.8256, 0.4199) | (0.8166, 0.1532) | (0.8591, 0.7181) |

Table 6 shows the granular rules for determining the data points' belonging. Here, $IG_y$ and $IG_n$ also represent the optimal granular intervals of data included and not included in a specific cluster, respectively. Comparing these results with that of Table 2, it is seen that granular intervals via the reconstruction error metric in Chameleon data have better distinguishing ability, which may be because of its more obvious spatial structures. Moreover, Cluster 4 and Cluster 6 perform the best among all the clusters; the other clusters keep some small fuzzy regions with uncertainty for online clustering.

**Table 6.** Results of granular intervals in the medium space.

|  | **Cluster 1** | **Cluster 2** | **Cluster 3** |
|---|---|---|---|
| $IG_y$ | [0, 0.022] | [0, 0.003] | [0, 0.006] |
| $IG_n$ | [0.0018, 0.3038] | [0.0387, 0.4747] | [0.0039, 0.2559] |
|  | **Cluster 4** | **Cluster 5** | **Cluster 6** |
| $IG_y$ | [0, 0.021] | [0, 0.005] | [0, 0.01] |
| $IG_n$ | [0.0300, 0.2203] | [0.0037, 0.2137] | [0.0509, 0.4409] |

Finally, with the combination of structural IGs in input space and granular rules in medium space, we could complete rule-based modeling as (15). Then, the online clustering on 1000 testing points is evaluated. The numerical results of performance metrics are presented below.

In Table 7, the results of several different online clustering on Chameleon data are presented. Firstly, for the proposed granule-based method, different amounts of granules are compared for discussion. By setting the same number of granules in each cluster, granules in the input and medium space are constructed for rule-based modeling, and are subsequently used to guide online clustering. From the comparative results, it is found that the accuracy of the proposed method on online clustering increases as more granules are used for modeling; meanwhile, the computation time also increases as well. Therefore, a trade-off should be made for selecting the optimal parameters in real applications. Secondly, by comparing with the two other DBSCAN extensions (i.e., C-DBSCAN and G-DBSCAN) on online clustering, it is found that the major superiority of the proposed method is less computation cost; this would be more obvious in big data analysis. Moreover, the accuracy of the proposed method can also outperform other DBSCAN methods when selecting suitable parameters. The reasons behind this phenomenon can be explained as: (1) C-DBSCAN leverages huge information of historical data in the online clustering process, causing a high computation cost; (2) G-DBSCAN can reduce the time consumption with the help of grid cells, but sub-grids have poor representation ability, especially for a dataset with a lot of anomaly data (e.g., blue points in Figure 10), leading to an unsatisfactory accuracy. The proposed granule-based DBSCAN method makes use of the granules' structure description ability and good representation ability, and both the time consumption and accuracy of online spatial data clustering are improved. Therefore, it is concluded that these results not only verify the feasibility of the proposed method on online clustering, but also illustrate

the improvement of the proposed DBSCAN extension on online spatial data clustering. Thirdly, comparing results with the other two *k*-means-based online clustering methods, the proposed method has great superiority on accuracy, not on computation time. This is because the Chameleon data has stronger constraints on the structure clustering, so conventional clustering methods, such as *k*-means, cannot perform well, even though they have superiority on computation time.

**Table 7.** Performance of online clustering on the Chameleon data.

|  | Granule-Based | | | | C-DBSCAN | G-DBSCAN | *k*-Means | W-*k*-Means |
|---|---|---|---|---|---|---|---|---|
|  | *c* = 5 | *c* = 10 | *c* = 15 | *c* = 20 | | | | |
| Acc. | 0.7550 | 0.8760 | 0.8990 | 0.9134 | 0.8979 | 0.8754 | 0.5218 | 0.5854 |
| Time | 0.0671 | 0.1185 | 0.1861 | 0.1907 | 0.4347 | 0.3924 | 0.1419 | 0.1576 |

All in all, by combining the results in both the toy dataset and the benchmark Chameleon dataset, we can conclude that it is feasible to apply the proposed DBSCAN-based granules to extend DBSCAN in online clustering. Moreover, the proposed method in this paper could achieve better accuracy on online spatial data clustering than conventional methods, and it can also have superior computation time to other DBSCAN extension algorithms on online clustering.

### 6. Conclusions

To realize fast and effective online spatial data clustering, this paper developed a DBSCAN extension algorithm with a combination of granular models focusing on online clustering. The proposed algorithm was developed with three layers via granular computing. The first layer is to construct structural granules via DBSCAN and GrC in the input space. The numerical results on synthetic spatial data verify the superiority of DBSCAN on clustering data with arbitrary shapes compared to conventional methods. Moreover, DBSCAN-based granules are found to well inherit the structure description ability, which is useful in spatial data clustering. Then, the second layer made use of the reconstruction error for rule formation in the medium space. Granular intervals are constructed via justifiable granulating, and utilized to form fuzzy rules for determining the data's belonging. The numerical results show that these rules have a good ability on distinguishing different clusters' data, and this ability will be outstanding when the data are more structural. Finally, making use of structural DBSCAN-based granules and fuzzy rules in medium space, online clustering on spatial data was implemented by rule-based modeling. Experiments on synthetic and typical spatial datasets verified the feasibility of the proposed method in online spatial data clustering. Through the comparison with well-known *k*-means-based methods, the superiority of the proposed method on improving the performance of spatial data clustering is verified. Moreover, via the comparison with different DBSCAN extensions on online clustering, the proposed granule-based method is demonstrated to have less computation overhead and a relatively good performance, especially in online clustering orienting to big spatial data.

By making use of this proposed method, researchers can develop some practical engineering applications about online recognition and spatial data in the future; for example, geographical information systems, logistics monitoring services, and so on. However, some limitations can also be found. For example, since the major superiorities of the proposed method are based on DBSCAN and GrC, it may be possible to develop online spatial data clustering algorithms based on more advanced density-based methods. Moreover, the discussion on how to trade off the number of granules with the computation overhead is not implemented in this paper, which would be also a meaningful topic in practical applications, which usually require comprehensive performances.

**Author Contributions:** Writing—original draft preparation, experiment, X.Z.; writing—review and editing, X.S.; writing and editing, supervision, funding acquisition, T.O. All authors have read and agreed to the published version of the manuscript.

**Funding:** This research was supported by JSPS KAKENHI Grant Number JP22K17961.

**Institutional Review Board Statement:** Not applicable.

**Informed Consent Statement:** Not applicable.

**Conflicts of Interest:** The authors declare no conflict of interest.

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
