# Peer review of "Extension of DBSCAN in Online Clustering: An Approach Based on Three-Layer Granular Models"

_applsci, doi:10.3390/app12199402_

Round 1

Reviewer 1 Report

In this work, the authors have developed a new extension of DBSCAN with the combination of granular models. First, by making use of DBSAN algorithms' advantages at extracting structural information, it becomes feasible to deal with nonlinear spatial datasets. Second, a series of granular models have been constructed through granular computing to guide the clustering. Experiments on a synthetic toy dataset and a typical spatial dataset validate these mentioned superiorities of the proposed clustering algorithm. Overall, the paper is somewhat interesting, and there are some new results in the paper. However, the paper needs minor revision before acceptance.

1.     The quality of the English language should be improved carefully with the help of a native English speaker to enhance readability.

2.     Please describe the motivation of Eq. (4).

3.     In Eq. (7), you have used an exponential function. Can we use other functions also?

4.     Summarizes the advantages and drawbacks of the suggested method in practical applications.

5.     A comparative study with some well-known methods is recommended.

6.     The conclusion section - The authors will have to demonstrate the impact and insights of the research. The authors need to provide several solid future research directions. Clearly state your unique research contributions in the conclusion section.

Reviewer 2 Report

There are many grammatical errors in this paper.

The simulation part is slightly thin.

Round 2

Reviewer 2 Report

The revision is OK.